# Canada as a Case Study for Balanced Presentation to Address Controversy on Emission Reduction Policies

**Robert V. Parsons**

I.H. Asper School of Business, University of Manitoba, Winnipeg, MB R3T 5V4, Canada;
Robert.Parsons@umanitoba.ca or robertvparsons@gmail.com

**Abstract:** Controversy is common on environmental issues, with carbon taxation in Canada a current example. This paper uses Canada as a case study for analysis based around balanced presentation, a technique developed some time ago, yet largely forgotten. Using the method, analysis is shifted away from the point of controversy to a broader quantitative question, with comparative data employed from official government sources. Simple quantitative analysis is applied to evaluate emission trends of individual Canadian provinces, with quantitative metrics to identify and confirm the application of relevant emission reduction policies by individual jurisdictions. From 2005 through 2019, three provinces show consistent downward emission trends, two show consistent upward trends, and the remaining five have no trends, showing relatively "flat" profiles. The results clarify, in terms of diverse emission reduction policies, where successes have occurred, and where deficiencies or ambiguities have existed. Neither carbon taxation nor related cap-and-trade show any association with long-term reductions in overall emissions. One policy does stand out as being associated with long-term reductions, namely grid decarbonization. The results suggest a possible need within Canada to rethink emission reduction policies. The method may be relevant as a model for other countries to consider as well.

**Keywords:** balanced presentation; emission reduction; carbon tax; cap and trade; carbon pricing; renewable fuel; energy efficiency; vehicle electrification; electricity decarbonization; grid intensity

## 1. Introduction

Controversy is a constant feature of environmental issues, with rational debate difficult when stakeholders hold strongly opposing positions. The public-at-large can be left uncertain as to who or what to believe, and can slip into cynicism, exactly what has been occurring within Canada on the subject of carbon taxation. On a national basis, Canada has been divided on the use of carbon taxation, including the related policy of cap-and-trade, as means to reduce greenhouse gas (GHG) emissions. However, the country also represents a microcosm of positions held and circumstances faced all over the world. As a geographically large, federated state, Canada at a sub-national (provincial) level exhibits varying circumstances in terms of energy and emission [1]. Further, Canada's ten main provinces, since 2005, have undertaken diverse policy approaches toward addressing emissions, with differing experiences and results.

Given this diversity, Canada represents a useful case study to address controversy on emission reduction policies. One possible approach is explored in this paper, based around balanced presentation. The technique was originally developed and applied in Canada by economist and statistician Stanley Warner [2], but today is largely forgotten. The objective of this paper is thus to employ a balanced presentation approach to consider emission reduction in Canada, using data from official government sources combined with simple quantitative analysis methods to compare GHG emission trends across Canadian provinces over fifteen years, from 2005 through 2019, and then to compare emission results to corresponding major reduction policy categories undertaken by the same individual





jurisdictions, with policy applications in each case confirmed using quantitative metrics. The analysis identifies policies that have been associated with long-term emission reduction, as well as situations where deficiencies or ambiguities exist. It is understood that emission reduction policies undertaken by governments can involve multiple objectives; however, they must at some point be able to demonstrate environmental effectiveness for reducing emissions.

The Background and Literature section outlines functional policy categories that have been considered for emission reduction, identifies evaluation criteria considered to determine policy success, describes general controversies surrounding carbon taxation, outlines the implementation of carbon taxation in Canada and associated controversies, and, lastly, describes the basic principles employed for balanced presentation. The Methods section describes how balanced presentation has been applied—in this case, specifically to assess emissions trends at a provincial level compared to corresponding implemented emission reduction policies—provides all data sources, and describes limitations of the work. The Results section presents long-term emission trends determined for individual jurisdictions, identifies reduction policy categories confirmed to have been consistently and sufficiently undertaken for each, and then considers how emission trends and policy categories correspond. The Discussion section outlines implications and future possible research directions. The intent is to yield insights, not just on whether or not carbon taxation itself may be useful in practice as undertaken in Canada, but to answer a broader question: "If not carbon taxes, then what policies can be undertaken to adequately reduce emissions?"

## 2. Background and Literature

### 2.1. Functional Policy Categories for Emission Reduction

There is a broad consensus that GHG emissions need to be adequately reduced on a worldwide basis, as outlined under the Paris Agreement [3]. Exactly how this can be completed, however, is not entirely certain [4], or uniform across different countries [5,6]. A variety of policies can be potentially applicable to reducing emissions, with one report alone identifying more than 50 diverse individual policies within just Canada [7]. Further, policies may be used in different combinations [8,9].

In terms of identifying relevant reduction policy categories, it is important to distinguish between broader climate-related policies that may involve adaptation, resiliency, or social equity aspects, versus policies intended to specifically reduce emissions. The latter are relevant in this case. Further, it is important to distinguish policies that are aspirational in nature versus those more functionally oriented. The former may include developing reduction strategies or reduction targets, including making pledges, providing better information, or implementing procedures for reporting or auditing. These may be useful for overall ultimate success, but have only indirect effects. Support of research, development, and demonstration (RD & D) is not included for similar reasons. While important for long-term progress, potential results from individual RD&D activities tend to be unpredictable. The focus is on specific policies anticipated to directly produce tangible reduction outcomes.

For the purposes of this paper, seven main functional policy categories are identified for emission reduction and include: (i) carbon taxation; (ii) cap-and-trade; (iii) grid decarbonization; (iv) renewable content in fuel; (v) electrification of vehicles; (vi) efficiency; and (vii) reducing fossil fuel development. These policy categories, as stated, primarily follow recent recommendations to the U.S. government [10], and are also consistent with other works over time [5,8,11–14].

Carbon taxation, of course, involves an economic instrument, specifically a form of emission-fee system. There is vast literature on this subject, with many background reviews outlining working principles [15,16], thus not requiring in-depth explanation here. Carbon taxation is closely related, but not identical, to cap-and-trade, which is sometimes termed emission trading systems or ETS, and is also an economic instrument. The objective for both is to reflect the cost of externalities—in this case, GHG emissions—in market transactions,

balancing benefits and costs in an attempt to reach some sort of rational equilibrium and, hopefully, stem pollutant release. Because of differences in operations, carbon taxation and cap-and-trade are here treated as separate policies. In the former, the price is set with the level of reductions determined by the resulting equilibrium, while in the latter, the required reduction is set with the price determined in the market by the trading of decreasingly available emission permits.

Additional background on the other policies is also relevant. Grid decarbonization involves reducing the GHG intensity of electricity, through a combination of enhancing renewable electricity sources and reducing generation based on coal or oil [17]. Transportation represents a significant source of emissions worldwide. As such, increasing renewable content provides a simple approach to decarbonizing conventional fuels, and has already been employed for some time [18]. The main options are ethanol, in the case of gasoline, and biodiesel or renewable diesel in the case of diesel fuel. Hydrogen also represents a potential renewable fuel option for the future, although it is still too nascent in terms of implementation to be relevant. Electrification of vehicles involves an advancing new technology for transportation as well, motivated primarily by the rapid development and commercialization of battery technologies. Such vehicles shift the source of motive energy from fossil fuels to electricity, with emission reductions depending on grid intensity. A transition to electric vehicles also offers efficiency gains such that emissions are generally lower, with reductions increasing as grids become more decarbonized. While offering potential, electric vehicle technology, however, is still at a relatively early stage of adoption in most parts of the world [19].

Improving efficiency to reduce emissions is a self-evident opportunity, and includes improving the energy efficiency of electrical appliances, buildings, motor vehicles, and industries, as well as improving the production efficiency of industrial processes. In terms of energy efficiency, recent work on EU countries shows a strong negative relationship between overall national energy efficiency, measured as annual gross domestic product (GDP) divided by annual gross domestic energy consumption, compared to annual total national GHG emissions, i.e., more efficient means lower emissions [20].

In terms of deliberate efforts to improve efficiency, vehicle fuel consumption standards have been common internationally, as well as some aspects of building and appliance efficiency improvement, but less so regarding industrial processes [14]. At the same time, deliberate efforts to improve energy efficiency have been seen to involve problems, especially in terms of measuring emission reductions and comparing the performance of efficiency efforts between jurisdictions. A well-known concern with energy efficiency involves "rebound" effects, reducing effectiveness over time [21]. Fundamental concerns also exist regarding an over-reliance on energy efficiency [22].

In terms of comparing the efforts of individual jurisdictions, the American Council for an Energy Efficient Economy (ACEEE) has provided "scorecard" comparisons at a national level between countries for some time. Concerns, however, exist with such "scorecard" approaches, given that metrics are known to embed assumptions, and thus potential biases, in their weightings. As such, independent rankings by one organization do not necessarily coincide with those of another. For example, ACEEE ranked Canada overall low on its 2018 International Energy Efficiency Scorecard, in a tie with the U.S. at #10 out of 25 countries assessed [23]. However, in early 2019, over largely the same timeframe, the U.S. Green Building Council ranked Canada as #2 in the world for LEED-certified building projects in 167 countries where the approach is applied [24]. The two involve overlap, but this is not reflected in their scores, appearing inconsistent. There is a general need for better metrics to assess deliberate energy efficiency efforts, especially relating to emission reductions.

The last category involves policies to move away from fossil fuel development. This is also a self-evident opportunity to reduce emissions, given that higher levels of fossil fuel extraction directly lead to higher production-related (upstream) and consumption-related (downstream) GHG emissions. While recognizing the desirable policy direction

indeed would be to de-emphasize fossil fuels [10], an important reality for Canada over the recent past has been increasing production. In 2005, annual Canadian primary energy production was approximately 16,500 PJ [25]. By 2019, this had increased to approximately 21,400 PJ [26], meaning a net increase on a national basis over the period of around 4900 PJ annually, roughly a 30% increase on an energy content basis over fifteen years. A further corollary of fossil fuel production is fugitive methane emissions. Methane represents a potent GHG, and a growing concern, especially in North America. While agriculture and natural sources remain significant, the fastest growth has been contributions from fossil fuel-related industries [27]. The latter sources of emissions also have been recently identified as significantly underestimated, including within Canada [28,29]. Being directly related to fossil fuel production, fugitive methane is thus considered here.

### 2.2. Success Evaluation Criteria

Policies to reduce emissions not only need to be first formulated, such as those described above, but also to be assessed. Four major criteria can be identified that have been considered for evaluation, including: (i) environmental effectiveness [5,8,30]; (ii) economic efficiency [5,30,31]; (iii) institutional feasibility [5,30,31]; and (iv) distributional implications [5,31]. Environmental effectiveness is the most obvious and essential criterion. If a policy is implemented with the expressed intent to reduce GHG emissions, actual emission reductions need to be demonstrated, desirably large reductions.

At the same time, evaluation of such policies has been multifaceted, involving the other three criteria. Economic efficiency considers the ability to provide net benefits, including indirect and competitiveness effects, or alternatively to show low costs per tonne of reductions. Institutional feasibility includes political acceptance, ease of implementation, data availability, adaptability, and consistency with other goals, including environmental. Distributional implications involve fairness of impacts and ensuring equity across societal groups. This last aspect appears to have become increasingly prominent in order to achieve a desired "double dividend" of reducing emissions and simultaneously enhancing social welfare [32].

An important technical aspect of economic efficiency relates to assessment methodologies, as employed. Guidelines from the UNFCCC [30] suggest the use of alternative scenarios to evaluate both emission reductions and net benefits. This obviously simplifies analysis and assessment—for example, allowing comparisons between scenarios, particularly "business as usual" scenarios. It also means, however, that environmental and economic modeling, with embedded assumptions, play a strong role in determining or even predetermining outcomes, as opposed to comparing changes in actual emission results as they occur over time. For example, recent work through the International Monetary Fund (IMF) relates reduction potentials for the Paris Agreement compared to 2030 "business as usual" scenarios for different countries, rather than any actual emission results [33].

### 2.3. Controversy Regarding Carbon Taxation

Carbon taxation is a prominent policy both employed and promoted internationally for the reduction of GHG emissions. Carbon taxation has many prominent advocates, including, for example, the International Monetary Fund [33,34] and World Bank, particularly through the Carbon Pricing Leadership Coalition. At the same time, however, carbon taxation remains controversial. Considering earlier identified policy evaluation criteria, the key strength of carbon taxation appears to be associated with economic efficiency [15,16]. On the other hand, weaknesses appear regarding two other evaluation criteria: institutional feasibility—specifically, political acceptance—and, most importantly for this paper, environmental effectiveness, whether adequate reductions can be achieved.

Political controversy with carbon taxation is well known, with the policy unpopular with voters in many countries. This includes prominent policy reversals in Australia and France, and referendum losses in the U.S. State of Washington [35–37]. Several recent papers explore the lack of popularity in more detail [38,39]. Although political perspective is noted

as an important factor in resistance, consumer response to pricing is relevant. Consumers generally agree with concepts such as "reduce emissions" or "polluter pay," particularly when no price is involved. However, acceptance of a discreet carbon tax drops off as consumers begin to be faced with the prospect of consequential prices, although also not linearly. This leads to seemingly contradictory results from within the same populations, expressing both support and distain. At the same time, this is not unexpected, given that such behavior is recognized in "willingness-to-pay" (WTP) evaluations as employed for new products in business. Consumer interest declines as the price increases, with responses in contingent valuation assessments depending on how survey questions are worded. Survey results and earlier WTP evaluation specifically for carbon pricing confirm that some cost level can be acceptable, but is relatively low [38,40]. Further, price levels that may be politically acceptable may not be sufficient to be effective, as discussed later.

The most important concern with carbon taxation is whether the approach can, in practice, demonstrate environmental effectiveness. Recent research shows that, despite prominence, surprisingly few evaluations have been undertaken on actual performance of carbon taxation schemes in place. Assessment results show significant performance limitations, with equivalent reductions in the range of only 0% to 2% annually, which is low, and with high variability in performance also exhibited [41]. At the same time, carbon taxation schemes appear overall to have performed somewhat better than cap-and-trade systems.

A long-identified concern regarding carbon taxation is that it cannot intrinsically guarantee emission reductions [5]. While strongly supported by economic theory, the effectiveness of carbon taxation is limited by economic principles. The mechanism of reduction involves increasing the price on a problematic fuel, which in turn makes that option less attractive, and thus reduces consumption and corresponding emissions [15,16]. The extent of reductions, however, depends on two factors: firstly, the level of the tax; and, secondly, the responsiveness of consumers to the imposition of the tax. Both introduce problems regarding environmental effectiveness.

Recent research shows that by 2020, carbon pricing levels in roughly two thirds of systems worldwide were only USD 20 per tonne or less [42], with even the World Bank's Carbon Pricing Leadership Coalition noting that pricing levels during the same time period remained substantially low [43]. This is problematic in the context of reaching Paris Agreement goals. Many researchers suggest that much higher prices will be required. Earlier work by the High Level Commission on Carbon Prices suggested that, by 2020, in the range of USD 40 to 80 per tonne and, by 2030, in the range of USD 50 to 100 per tonne would be required [44]. Recent work by the IPCC suggests that higher prices on emissions are necessary, with inferred minimum values of more than USD 150 per tonne by 2030 and more than USD 250 per tonne by 2050 [13]. Such price levels, however, are potentially damaging to political acceptability. As such, overcoming political resistance has become a major component of carbon tax development and associated discussions [45].

Consumer responsiveness, or lack thereof, is also a concern. A characteristic of commodity fossil fuel markets observed across the economic literature is inelasticity [46], as described technically using the price elasticity of demand. Consumers appear to have become more inured and less responsive to fuel price changes over time [47,48]. This has been further exacerbated by price volatility [49]. Such trends can help to explain the apparent lack of effectiveness of carbon taxation. Price elasticity of demand, however, is known to be complex [50], with regional variability [51], and with declining consumer responsiveness not necessarily applicable in all circumstances [52]. Importantly, consumer responsiveness cannot be taken for granted. When considering the pricing effects of carbon taxation, the tax level acts as a "price signal" rather than a carbon cost per se. When consumer responsiveness is low, the effective cost per tonne for reductions is increased, directly impacting both environmental effectiveness and economic efficiency.

## 2.4. Carbon Taxation in Canada and Associated Controversy

Canada's current federal government came to power in 2015, in part based on an apparently pro-environmental platform, promising, as its marquee policy, to implement a national price on carbon. However, a national pricing structure was not actually implemented until into 2019, by which time opposition had become much more vocal. Canadians became polarized and deeply divided on this issue, potentially contributing to the result of the federal election in the fall of 2019, which reduced the government to a minority status. Diverse opinions have been expressed within Canada on carbon taxation, both in favor [53–56] and opposed [57–60], with opposition arguments involving concerns relating to both political acceptance and environmental effectiveness.

The federal government, however, became engaged in emission reduction as a priority policy much later than sub-national governments. Diverse emission reduction policies had been implemented by provincial governments over the decade prior. These included carbon taxation, cap-and-trade, and other measures.

Selection of carbon taxation at a national level was significantly attributed to the earlier apparent success in one province, namely British Columbia [61], which had implemented its own carbon tax in 2008, a measure that was, at the time, pioneering [56]. The federal policy model substantially followed British Columbia, with the implication that all carbon taxes within Canada involve consistent structure and application, including: broad-based levies on commodity fossil fuel products according to GHG content [62], with these levies generating the overwhelming proportion of tax revenue [63]; increasing levy amounts over time; a separate output-based pricing system (OBPS) for selected large emitters, not discussed further; and revenue recycling back to consumers.

Federal legislation in Canada set minimum requirements for carbon taxation, or an equivalent level if opting for a cap-and-trade system. Price levels were initially set at USD 16 per tonne in 2019, rising by USD 8 per tonne annually, to a level of USD 40 per tonne in 2022 [62]. The decision to require provinces to meet stated tax levels, or face imposition of a federal backstop, rather than emission reductions, led to a variety of inconsistencies. These included, in some cases, the deeming of renamed existing fuel taxes as sufficient [64], and, in other cases, charging carbon taxes on elevated renewable fuel content.

The federal system has provided revenue recycling, as was done in British Columbia, but not quite in the same form, specifically employing a rebate system primarily directed to individual households [65]. This involved unit rebates per person that decline for each additional household member, but with no income testing. Initially, this approach was claimed to be for enhancing social equity, addressing the fourth success criterion. The nature of the federal rebate system, however, appears more directly tied to addressing political acceptance [66]. Questions arise in terms of social equity aspects. It was claimed that a majority of families would gain more back than paid in taxes, and with lower-income families benefiting the most [67]. However, financial results for the carbon tax in 2019 showed that households paid significantly more in carbon tax than rebates on average, with total tax revenues being higher than anticipated, and total rebates lower than anticipated [63]. Further, there has been no reporting of how rebates were actually distributed on an income basis, in order to be able to confirm social equity aspects.

The performance of Canada's federal carbon tax, so far, is also problematic in terms of environmental effectiveness, particularly when compared to the high expectations assigned to the tax. A uniform tax level of USD 16 per tonne, or equivalent for cap-and-trade, was required uniformly across the country for the first time during 2019, as noted, including the backstop pricing system applied to four provinces deemed to be non-compliant. However, despite a carbon price being fully in place, emissions in 2019 went up from 2018, not down [68]. Increased use of transportation fuel was specifically identified as an area of concern, with such fuels being a particular target for the carbon tax. Canada's federal government had been counting on large, decisive emission reductions over a short period of time, namely 80 million to 90 million tonnes by 2022, based solely on this measure alone [61]. This anticipated reduction is compared to a "business as usual" scenario rather

than actual emissions, but represents in the range of 11% to 12% of Canada's total emissions for 2019. Further concerns have recently come to light regarding British Columbia's carbon tax. Earlier analyses had suggested the tax to be effective for reducing emissions [53–55], with such studies used in part to justify the federal policy [61]. Recent research on British Columbia, however, suggests that while transportation sector emissions were reduced somewhat, there have been no overall reductions in that province [41,69].

Although Canada was not specifically singled out for criticism as part of the 2019 Emissions Gap Report [70], the country, as one of the G20, which produce most of the emissions in the world, was noted along with others as unlikely to meet its existing commitment for the Paris Agreement by 2030. The report further emphasizes a need to achieve 7% to 8% annual reductions if targets are to be achieved, a level that Canada has never approached. The response in 2020 was a new plan, one that included further increases in the required carbon tax level up to approximately USD 140 per tonne by 2030 [71], and in 2021 to further increase reduction targets [72]. A later report by the Parliamentary Budget Officer, however, suggested that the tax level likely would need to be much higher, closer to USD 200 per tonne by 2030 [73]. More recent research suggests that Canada has a low probability of achieving its Paris Agreement target based on nationally determined contributions, despite increasingly ambitious goals being stated [74].

*2.5. Balanced Presentation Technique*

Canada has failed to make progress on emission reductions and is relying on the success of the carbon tax. At the same time, controversy has not abated, making it difficult to find a rational pathway forward.

On the other hand, given diversity, options exist to find better way to undertake objective discussions. One potential approach is based around balanced presentation. The technique was originally developed and applied in Canada by economist and statistician Stanley Warner [2]. Warner directly used the balanced presentation technique to consider two issues in the mid-1970s and early 1980s. These specifically involved express roadway construction [75] and airport expansion [76]. Both cases involved controversies, and in both cases, environmental concerns were central. Warner's findings were valid and borne out, based on historical follow-up.

The objective of balanced presentation is to ensure that positions on a policy issue can be fairly and adequately represented. To accomplish this, Warner's approach was not overly prescriptive. Instead, it involved two key principles from an economics and policy perspective that are simple, yet sound, and powerful in helping to move discussions forward. The first principle involves shifting the question away from the point of controversy itself to one still relevant but quantitative in nature, and thus more bland and amenable to objective review. For illustration, the obvious direct question for this paper would be to ask, "Is a carbon tax the best way to reduce GHG emissions?" This question was indeed posed as part of a 2019 online discussion between two Canadian academics, one an advocate, the other more skeptical [77]. While their discussions were civil, both were passionate and resolute, with no minds changed on the subject. It is thus advantageous to consider questions that are less directly controversial.

The second principle involves ensuring that information relevant to different perspectives is available through single sources, and not hidden. There is a tendency to only present information supportive to a cause, and not present what could suggest a more nuanced situation. Warner also employed sophisticated statistical analysis, but with this used to assess and ensure the objectivity of the information sources. If legitimately singular and objective sources of information are already available—for example, official emissions release data—the latter becomes less necessary.

Mindful of Warner's suggested first principle when considering carbon taxation, it is possible to formulate a related but broader quantitative question for analysis:

> "What have been the actual trends in total GHG emission releases from different provinces across Canada since 2005, based both on total annual emissions

and percentage emissions relative to 2005 levels, and how, if at all, have major emission reduction policy categories employed within those individual Canadian jurisdictions been associated with respective emission trends?"

This question does not directly discuss carbon taxation and is thus intrinsically less controversial. Nevertheless, the question permits useful comparative analysis. The question focuses on environmental effectiveness, which, as discussed, is the most important evaluation criterion in determining policy success. The question also considers comparative performances of different provinces, relative to 2005, which Canada has used as its baseline for the Paris Agreement target. As discussed earlier, carbon taxation was adopted based on its apparent success in one province, and the question capitalizes on the fact that sub-national governments led on exploring diverse emissions reduction policies over a long period before the federal government became prominently engaged.

All necessary GHG data for Canada and individual provinces by year from 1990 are available from one single source, namely Canada's annually released National Inventory Reports. Such reports constitute official submissions to the United Nations Framework on Climate Change, with a consistent two-year reporting delay. In terms of relevant provincial policies applied to emission reduction, all governments report their policies, but not necessarily in an objective manner. It is nevertheless possible to use quantitative metrics to confirm consistent and sufficient implementation of relevant policy categories in order to be counted, with the data for metrics also available via singular, official sources from the government.

## 3. Methods

The method employed in this paper was based on using the principles associated with balanced presentation, and involved two analysis components: (i) evaluating emission trends for all of Canada's ten provinces over the period from 2005 through 2019, based on total annual emissions and percentage emissions relative to 2005; and (ii) evaluating and confirming major emission reduction policy categories in play within the same jurisdictions over the same period. This permits comparison of implemented policies in each case to the corresponding emissions trend of each jurisdiction.

To evaluate emissions trends, data for all provinces, as noted, are available and released as part of Canada's annual National Inventory Reports, with the most recent version in 2021 covering emissions up to 2019 [78]. Importantly, minor adjustments can be, and have been, made in past reports, such that only the most up-to-date data were employed, being available as a complete data set. Data set: GHG_Econ_Can_Prov_Terr.csv available online: https://open.canada.ca/data/en/dataset/779c7bcf-4982%E2%80%9347 eb-af1b-a33618a05e5b (accessed on 21 April 2021).

The resulting 15 years of data were evaluated using a simple Microsoft Excel spreadsheet to determine whether, and to what extent, emissions have changed. As such, the focus was on environmental effectiveness in terms of overall emission reduction. Simple linear regressions were considered for the data of each jurisdiction, in order to assess: firstly, whether any linear trend was reasonably evident; and secondly, whether the resulting trend, if present, was upward or downward. The quantitative threshold to establish a trend was based on achieving a coefficient of determination ($r^2$) value of at least 0.70. Where reasonable trends were evident, the slope value, the $r^2$ value, and whether the trend was upward or downward are reported. Where a sufficient trend threshold was not achieved, this result is reported, also reporting the mean annual emissions value over the timeframe and the coefficient of variation (CV), i.e., standard deviation (SD) divided by the mean.

In terms of policy directions of the individual jurisdictions, the same seven major emission reduction policy categories as earlier identified were employed, with a slight change in the last category: (i) carbon taxation; (ii) cap-and-trade; (iii) grid decarbonization; (iv) renewable content in fuel; (v) electrification of vehicles; (vi) efficiency; and (vii) fossil fuel development. The change in the last category reflects the earlier identi-

fied reality that fossil fuel production over the period increased, and thus characterizes a counter-effective activity.

To ensure greater objectivity in the assignment of whether polices may have been applicable to individual provinces, quantitative thresholds were applied to each category, as outlined in Table 1.

**Table 1.** Quantitative thresholds used to confirm applicability of functional policy categories.

| Item | Major Policy Category | Quantitative Threshold |
|:---:|:---:|:---:|
| 1 | Carbon taxation | Measures in place more than three years with broad-based tax, and increasing price levels |
| 2 | Cap-and-trade | Measures in place more than three years with broad-based fossil fuel limits, trading and goals set |
| 3 | Renewable content in fuel | Renewable content ≥ 2.5% above federal minimum level for more than three years |
| 4 | Grid decarbonization [1] | Grid intensity reduction ≥ 125 g per kWh on a generation basis from 2005 to 2019 |
| 5 | Vehicle electrification | Registered light-duty vehicles reaching ≥ 2.5% having plug-in capability |
| 6 | Energy efficiency | No adequate or consistent metrics yet available |
| 7 | Fossil fuel development [2] | Increase of ≥ 500 PJ in energy content of annual primarily fossil fuel energy production from 2005 to 2019 |

[1] Overall grid intensity for Canada as a whole declined 100 g per kWh on a combustion basis from 2005 to 2019 [78], with the defined threshold set as 25% higher than the aggregate level. [2] Overall energy content of primary production in Canada increased by around 4900 PJ from 2005 to 2019 [25,26], with the defined threshold representing more than 10% of the national increase.

The intent of using thresholds was to confirm that policy measures were consistently and sufficiently applied, particularly given that there had been a tendency toward policy "dabbling" by many provinces. The use of quantitative values also helped to ensure that circumstances, as differentiated, were statistically distinct.

The thresholds in most cases are self-explanatory, with some additional clarifications for three. Regarding electricity, data also from the National Inventory Report show that virtually all provinces reduced their grid intensity over the period from 2005 to 2019, with the aggregate national reduction being approximately 100 g per kWh on a combustion basis [78]. An elevated threshold was thus selected being at least 25% higher. On a generation basis, this translated to at least 125 g per kWh reduction.

Regarding deliberate efficiency efforts, as noted in Table 1, no suitable metrics were available to compare the performances of individual provinces. Ongoing international comparisons by ACEEE were noted earlier. Within Canada, however, a new national organization, Efficiency Canada, recently began providing "scorecard" rankings, broadly similar to ACEEE, for individual provinces in 2019 [79], thus not permitting historical comparisons covering the period.

Under the last category, regarding fossil fuel development, the threshold was set as a minimum net increase from 2005 to 2019 of 500 PJ annually, on an energy content basis. This represents more than 10% of the net increase on a national basis over the period, which was approximately 4900 PJ, as described earlier. Three major categories of fossil fuels are involved, with threshold values translated respectively to units more typically employed in each case, based on standard energy content values [80], as follows:

- For crude oil, ≥225,000 barrels per day (BPD) equivalent production increase, based on energy content of 6100 MJ per barrel equivalent;
- For natural gas, ≥1.27 billion cubic feet (BCF) per day equivalent production increase, based on energy content of 1.08 MJ per cubic foot;
- For metallurgical coal, ≥17 million tonnes annual production increase, based on energy content of 29,300 MJ per tonne.

Quantitative production data are publicly available from consistent government sources for all three major fossil fuel categories: crude oil, natural gas, and metallurgical coal [81–84]. Only metallurgical coal is considered in this case, given that implications for

thermal coal production and consumption are effectively covered separately under the "grid decarbonization" metric, thus avoiding double counting.

The last aspect of the Methods is to identify limitations. The analysis, as undertaken in this case, permitted comparison of emission trends to corresponding emission reduction policy categories for individual jurisdictions. This provided at least a preliminary understanding of whether or not individual policies have been associated with long-term emission reductions. The applicability of individual policy categories to individual jurisdictions was based on quantitative metrics; however, the analysis essentially treated individual policies on either a "present" or "absent" basis. This does limit further quantitative characterization, including proportional contributions from different policies. The analysis also focuses exclusively on the criterion of environmental effectiveness, with aspects relating to economic efficiency, institutional feasibility, and distributional implications not considered. These are important, although the ultimate judgment as to whether a jurisdiction can fulfill Paris Agreement commitments will be based on the extent of emission reduction.

## 4. Results

### 4.1. GHG Emission Trends for Individual Jurisdictions

The results from the analysis of overall emission trends are presented in Table 2, covering the period from 2005 through 2019, for each province and for Canada as a whole. Over the fifteen years, three provinces exhibited consistent downward trends in total emissions, two provinces exhibited consistent upward trends in total emissions, and five provinces exhibited no trends over the period. Within Table 2, the three provinces with downward trends are listed first, in order of $r^2$ value, reflecting the strength of relationships. The two provinces with upward trends are listed second, again in order of $r^2$ value. Lastly, the five remaining provinces, showing no clear trends, are listed in order based on CV value (lowest to highest), reflecting the relative flatness of respective emission profiles.

**Table 2.** GHG emission trends for individual jurisdictions over period from 2005 through 2019.

| Jurisdiction (Abbrev) [1] | Slope (MT/y/y) [2] | $r^2$ Value | Emission Trend | Mean (MT/y) [3] | CV (%) |
|---|---|---|---|---|---|
| NB | −0.60 | 0.93 | Downward | n/a | n/a |
| NS | −0.58 | 0.89 | Downward | n/a | n/a |
| ON | −3.09 | 0.76 | Downward | n/a | n/a |
| SK | +0.69 | 0.81 | Upward | n/a | n/a |
| AB | +2.91 | 0.77 | Upward | n/a | n/a |
| Canada | n/a | 0.06 | No trend | 722.4 | ±2.04% |
| QC | n/a | 0.44 | No trend | 82.6 | ±3.45% |
| BC | n/a | 0.06 | No trend | 61.7 | ±3.58% |
| MB | n/a | 0.45 | No trend | 21.2 | ±4.72% |
| NF | n/a | 0.26 | No trend | 10.6 | ±4.76% |
| PE | n/a | 0.60 | No trend | 1.9 | ±8.18% |

[1] Abbreviations: AB = Alberta; BC = British Columbia; MB = Manitoba; NB = New Brunswick; NF = Newfoundland and Labrador; NS = Nova Scotia; ON = Ontario; PE = Prince Edward Island; QC = Quebec; SK = Saskatchewan. [2] Slopes expressed in units of million tonnes (megatonnes) per year increase or decrease. [3] Mean annual emissions over period expressed in units of million tonnes (megatonnes) per year.

The provinces with no clear trends all show $r^2$ values significantly lower than 0.70, except one (i.e., Prince Edward Island at 0.60). The analysis thus appears to reasonably differentiate jurisdictions in terms of whether a trend is present or absent. Canada itself also shows no overall trend in emissions, being effectively flatter, in relative terms, than any individual province alone.

Results of emissions relative to 2005 levels, important in understanding progress towards Canada's Paris Agreement target, are presented in Table 3 for each province, and for Canada as a whole. The order of presentation in this case is from lowest to highest relative emissions percentage. For jurisdictions showing no trend, the relative levels for mean values from Table 2 are also included.

**Table 3.** GHG emissions relative to 2005 levels for individual jurisdictions.

| Jurisdiction (Abbrev) | 2019 Emissions Relative to 2005 (%) | Emissions Trend (Table 2) | Mean Emissions Relative to 2005 (%) |
|---|---|---|---|
| NB | 62% | Downward | n/a |
| NS | 70% | Downward | n/a |
| ON | 79% | Downward | n/a |
| PE | 86% | No trend | 92% |
| QC | 96% | No trend | 94% |
| Canada | 99% | No trend | 98% |
| BC | 104% | No trend | 98% |
| NL | 105% | No trend | 101% |
| MB | 110% | No trend | 102% |
| SK | 110% | Upward | n/a |
| AB | 117% | Upward | n/a |

While some provinces show demonstrable downward trends, Canada's overall emissions clearly have not been dropping, in 2019 being still at 99% relative to the national result for 2005. Reduction results for 2019 relative to 2005 show that the two leading provinces, New Brunswick and Nova Scotia, have already achieved Canada's 30% reduction target for the Paris Agreement. Ontario, the third leading province, also shows a reduction that is significant. Of the two provinces showing increasing trends, Alberta in 2019 reached 117%, while Saskatchewan reached 110%. Of the three provinces with the flattest profiles, no consequential reductions are seen, with, in 2019, Quebec reaching 96%, British Columbia reaching 104%, and Manitoba reaching 110%, just slightly lower than Saskatchewan. The remaining two provinces show the highest variability, with, in 2019, Newfoundland and Labrador reaching 105%, and Prince Edward Island, the most variable, reaching 86%.

*4.2. Emission Reduction Policy Categories Applicable for Individual Jurisdictions*

Using the quantitative thresholds outlined in Table 1, emission reduction policy categories identified as applicable to individual provinces over the fifteen-year period are summarized in Table 4, with individual policy categories discussed in more detail.

**Table 4.** Summary of emission reduction policy categories identified as applicable to jurisdictions.

| Emission Reduction Policy Category | Applicable Jurisdiction(s) |
|---|---|
| Carbon taxation | BC |
| Cap-and-trade | QC |
| Renewable fuel content | MB, SK |
| Grid decarbonization | AB, ON, NS, NB |
| Vehicle electrification | No jurisdiction applicable |
| Energy efficiency | No suitable measure for evaluation |
| Fossil fuel development | AB (#1) regarding crude oil BC (#2) regarding natural gas |

4.2.1. Carbon Taxation

Carbon taxation and cap-and-trade involve operational differences and are treated as separate policy categories. They are conceptually similar, and a common aspect of the threshold, as defined, requires consistent application and active involvement for more than three years. As such, the two policies are relevant only to one province each. For carbon taxation, this uniquely is British Columbia. As noted earlier, British Columbia first implemented its carbon tax in 2008, thus covering 12 of 15 years. Manitoba is noted to have implemented a small but targeted tax on remaining coal use, but this was trivial, and phased out [56]. Alberta implemented a broad-based carbon tax in 2017, but it was in place for less than three years, being repealed part way through 2019 [85]. Carbon taxation schemes in Newfoundland and Labrador and Prince Edward Island, as well as the federal

carbon backstop pricing system, as applied to Manitoba, New Brunswick, Ontario, and Saskatchewan, were implemented in 2019, too recent to be relevant.

### 4.2.2. Cap-and-Trade

As with carbon taxation, cap-and-trade is only relevant to one province—in this case, Quebec. Quebec began implementing a small carbon tax on certain fuels in 2007, but in 2013, it transitioned to a cap-and-trade system operated in collaboration with the Western Climate Initiative, based in California [56]. As such, Quebec's involvement with cap-and-trade has been extensive, covering at least 7 of 15 years. Ontario joined briefly with Quebec in the same system starting in 2017, but exited in 2018 [86]. Lastly, Nova Scotia also opted to employ a cap-and-trade system, but only began in 2019. The timeframes for the latter two jurisdictions are too short to be relevant.

### 4.2.3. Elevated Renewable Content in Fuel

This policy category, involving elevated renewable content in fuels, is relevant to two provinces, Manitoba and Saskatchewan. Both have been at least 2.5% above federal requirements, and both have had mandates in place since 2007, thus covering 13 of 15 years [87]. The current federal government has considered an enhanced Clean Fuel Standard, but this has not yet been implemented and thus is not relevant.

### 4.2.4. Grid Decarbonization

This policy category involves reducing the GHG intensity of the electricity grid, most notably achieved through the reduction or elimination of coal-based or oil-based generation. Quantitative data for all provinces are presented in Table 5 [78]. Generation-based grid intensity values are presented for each province for 2005 and 2019, respectively. As illustrated, grid decarbonization is relevant to four provinces: Alberta, Ontario, Nova Scotia, and New Brunswick. It is also important to note that, while not illustrated, calculations using consumption-based grid intensity provide identical results, in that the same four provinces achieved a reduction at least 25% more than aggregate.

**Table 5.** Quantitative comparison of generation-based grid intensities from 2005 to 2019.

| Jurisdiction (Abbrev) | 2005 Intensity (g/kWh) | 2019 Intensity (g/kWh) | Change (g/kWh) | Threshold Achieved | Reduction (%) |
|---|---|---|---|---|---|
| Canada | 220 | 120 | −100 | n/a | −45% |
| AB | 910 | 620 | −290 | Yes | −32% |
| ON | 230 | 30 | −200 | Yes | −87% |
| NS | 880 | 710 | −170 | Yes | −20% |
| NB | 400 | 260 | −140 | Yes | −35% |
| SK | 780 | 660 | −120 | No | −15% |
| PE | 100 | 2 | −98 | No | −98% |
| MB | 9.7 | 1.2 | −8.5 | No | −88% |
| BC | 25 | 18.6 | −6.4 | No | −26% |
| QC | 3.9 | 1.2 | −2.7 | No | −69% |
| NF | 20 | 27 | +7.0 | No | +35% |

Alberta had the highest grid intensity within Canada in 2005. In 2015, measures were implemented to begin significantly reducing coal-based generation, by 2019 reducing their grid intensity more than 30% and achieving the largest absolute reduction. Ontario has taken an aggressive approach, achieving a complete phase-out of coal-based generation by 2014 [88], and reducing their grid intensity by more than 85%. Nova Scotia reduced its grid intensity by more than 20%, primarily from coal reduction [89], while New Brunswick reduced theirs by more than 35%, primarily from oil reduction, with a significant factor there being the refurbishment and restart of their nuclear power plant [90].

### 4.2.5. Vehicle Electrification

In terms of vehicle electrification, Quebec has led Canada and, by 2019, more than 40% of such vehicles in Canada were concentrated within that province [91]. However, even in Quebec, total uptake is too low overall to be relevant, in the range of only 1% to 2% of registered vehicles. Thus, while electrification of vehicles may be a potentially useful measure to reduce emissions, vehicle numbers have not been sufficient to verify its effectiveness for overall emission reduction.

### 4.2.6. Energy Efficiency

As discussed in the Methods section, there are no consistent or adequately objective quantitative metrics available for the time period to allow for a comparison between provinces. Further comments are provided in the Discussion section.

### 4.2.7. Fossil Fuel Development

The final policy category is fossil fuel development. Quantitative results are presented in Table 6, broken down into three major components: crude oil, natural gas, and metallurgical coal. Two provinces show large increases in the primary energy production associated with fossil fuels, namely Alberta, based on crude oil production, and British Columba, based on natural gas production.

**Table 6.** Quantitative comparison of fossil fuel production energy content from 2005 to 2019.

| Fossil Fuel | Jurisdiction | 2005 Level [1] | 2019 Level [1] | Change | Threshold [2] |
|---|---|---|---|---|---|
| Crude oil | AB | 1,539,614 | 3,434,856 | +1,895,242 | Yes |
| " | SK | 419,009 | 487,056 | +68,047 | No |
| " | MB | 13,999 | 44,116 | +30,117 | No |
| " | ON | 2418 | 514 | −1905 | No |
| " | BC | 30,468 | 16,125 | −14,343 | No |
| " | NF | 304,763 | 261,698 | −43,065 | No |
| Natural gas | BC | 2.62 | 4.55 | +1.93 | Yes |
| " | ON | 0.02 | 0.01 | −0.01 | No |
| " | SK | 0.70 | 0.39 | −0.31 | No |
| " | NS | 0.39 | 0.00 | −0.39 | No |
| " | AB | 13.22 | 11.67 | −1.56 | No |
| Metallurgical coal | BC | 26.7 | 27.4 | +0.7 | No |

[1] Production levels listed for 2005 and 2019 are: barrels per day (BPD) for crude oil; billion cubic feet (BCF) per day for natural gas; and million tonnes per year for metallurgical coal. [2] Threshold values are: $\geq$225,000 BPD for crude oil; $\geq$1.27 BCF per day for natural gas; and $\geq$17 million tonnes annually for metallurgical coal.

Important in considering any changes in crude oil vs. natural gas is that in both Alberta and British Columbia, the production of one increased significantly while the other dropped. In both cases, however, the overall increase in fuel production is more than sufficient to cover the decline in the other, with an increase in total production of at least 500 PJ annually in both cases. With respect to crude oil, Canada as a whole, since 2005, has seen significant growth in production, with the overwhelming majority associated with oil sand-based deposits in Alberta. In many other provinces, on the other hand, crude oil has been in decline, including in British Columbia, Ontario, and Newfoundland and Labrador. Production levels did increase somewhat in both Saskatchewan and Manitoba, but they were not sufficient to meet the threshold.

With respect to natural gas, since 2005, Canada's overall production has remained relatively constant. The situation in Canada involves a shift in where production is occurring. Increased production in British Columbia has been greater than the threshold, and relevant. A reduction in natural gas production has occurred in other provinces, most significantly in Alberta. With respect to metallurgical coal, the only province involved significantly in this activity has been British Columbia, but the relatively small increase in production is not relevant.

Canada's overall increase in primary annual energy production from fossil fuels over the 15 years, as noted earlier, was approximately 4900 PJ. From the analysis, Alberta's net contribution is approximately 3600 PJ, or roughly 73% of the increase. British Columbia's net contribution is approximately 730 PJ, or roughly 15% of the increase. The contribution from all of the other provinces, combined, is thus only around 12%. While British Columbia's contribution is notably smaller than that of Alberta, it is larger than the identified threshold of 10%, and is also larger than all remaining provinces combined.

*4.3. Relationships of Implemented Reduction Policy Categories with Emission Trends*

The emission trends and identified relevant emission reduction policy categories for each jurisdiction are summarized in Table 7. Provinces are outlined in the same order as in Table 3. This order is based on emissions levels relative to 2005, from the lowest value to the highest value.

**Table 7.** Comparison of emission trends with identified reduction policy categories by province.

| Jurisdiction | GHG Emission Trend | Reduction Policy Categories |
| --- | --- | --- |
| New Brunswick | Downward | Grid decarbonization |
| Nova Scotia | Downward | Grid decarbonization |
| Ontario | Downward | Grid decarbonization |
| Prince Edward Island | No trend | No major policy identified |
| Quebec | No trend | Cap-and-trade |
| British Columbia | No trend | Carbon taxation, Fossil fuel production (#2) |
| Newfoundland and Labrador | No trend | No major policy identified |
| Manitoba | No trend | Renewable fuel content |
| Saskatchewan | Upward | Renewable fuel content |
| Alberta | Upward | Grid decarbonization, Fossil fuel production (#1) |

Results presented in Table 7 illustrate the associations between applicable policies and emission trends. Of the seven policy categories originally considered in this paper, grid decarbonization is the one strongly associated with long-term emission reductions, albeit not uniformly. All three of the top provinces, New Brunswick, Nova Scotia, and Ontario, can be quantitatively acknowledged as engaging in this policy. Alberta is the only exception, engaging in the policy but with no emission reduction seen.

Three policy categories appear to be not associated with any long-term emission reductions. Notably, these include both carbon taxation and cap-and-trade, but also elevated renewable fuel content. Quebec, which has been uniquely involved with cap-and-trade, shows the flattest emission profile, with no long-term decline. British Columbia, which has been uniquely associated with carbon taxation, shows the second flattest emission profile, with no long-term decline. Manitoba and Saskatchewan have been both involved with elevated renewable content in fuels. However, Manitoba shows the third flattest emission profile, while Saskatchewan's increased.

Two policy categories, efficiency and vehicle electrification, were not considered further for comparison. In the case of efficiency, provinces could not be evaluated given a lack of suitable quantitative data for comparison, while, in the case of vehicle electrification, no province could yet meet the quantitative threshold. The final policy category, fossil fuel development, characterizes counter-effective activities, and was confirmed, quantitatively, as emphasized in two provinces, Alberta and British Columbia. Alberta showed the largest increase in fossil fuel production and also showed the highest proportional increase in emissions. British Columbia showed the second largest increase in fossil fuel production while overall emissions were flat, with no long-term decline.

## 5. Discussion

The analysis, as undertaken, used Canada as a case study regarding the use of balanced presentation. The method was based on applying the principles of the technique and led to examining trends in official GHG emissions data for individual provinces and employing

quantitative metrics to identify and assign relevant reduction policy categories. While the initial motivation was to consider carbon taxation, a controversial topic within Canada and elsewhere, broader insights emerged. The results are relevant to Canada, but also highlight balanced presentation as a useful approach to consider for other countries.

One policy category stood out from the analysis as being associated with practical success in the long-term reduction of emissions, namely grid decarbonization. Such positive performance is also mirrored internationally, most notably by the United Kingdom. Success in the U.K. has been attributed to grid decarbonization, particularly the almost complete shut-down of coal generation [92]. At the same time, it is important to note that Alberta engaged in grid decarbonization, but the policy appears in this case to have been insufficient to stem or reduce overall emissions. On the other hand, a lack of adequate grid decarbonization appears to be a contributing factor in the continued increase in emissions seen within Saskatchewan. Saskatchewan remains one of the highest-intensity grids in Canada, and over the 15-year period showed the lowest proportionate decrease in grid-intensity, only 15%.

The positive outcome associated with grid decarbonization suggests the importance of a concept within Canada termed the "East–West" grid. This involves better using clean-energy resources internally within the country, rather than exporting to the U.S. [93]. This is not a new idea, but one seeing little progress so far [94], with obvious barriers and constraints [95]. Nevertheless, opportunities and benefits could be significant.

Fossil fuel development can be identified as a reason behind the continued increase in emissions observed for Alberta. High activity in British Columbia can be seen as a contributing factor for no emission reductions being seen in that province. These results tend to confirm a need to de-emphasize fossil fuel production if reductions are to be achieved. The use of elevated levels of renewable content in fuels, while intuitively a positive policy direction, also appeared to be not associated with any appreciable long-term emission reductions, based on experience in both Manitoba and Saskatchewan. Neither the provinces of Newfoundland and Labrador, nor Prince Edward Island, were identified with any policy category. These two provinces show no trend, and also exhibit the most variable data of all the provinces showing no trend.

Electrification of vehicles, as outlined, was not sufficiently implemented anywhere to meet the defined threshold, and was not assigned as an applicable policy category in any jurisdiction. Further investigation is warranted, particularly as electric vehicle registrations begin to reach sufficient levels, in order to determine if the approach could make a significant practical impact on emissions. Efficiency, especially energy efficiency, was not assigned as a relevant policy category, given a lack of data. In this case, however, there are further concerns that need to be addressed. As discussed in the Background and Literature section, there is already a lack of consistency when relying on "scorecard" rankings at a national level. Unfortunately, discrepancies also arise between recent internal rankings within Canada by Efficiency Canada [79], when compared with the results of this analysis. While New Brunswick, Nova Scotia, and Ontario achieved the top emission reductions by 2019, these same provinces are ranked by Efficiency Canada in the same year as #8, #4, and #3 on energy efficiency. The top two provinces named by Efficiency Canada, British Columbia and Quebec, on the other hand, show the flattest emissions profiles within Canada, with neither achieving any significant emission reductions. This points to a need for better metrics to assess energy efficiency, especially for emission reductions.

Lastly, this leaves consideration of carbon taxation, and related cap-and-trade. As noted previously, Quebec and British Columbia show the flattest emission profiles of all the provinces over the timeframe. Neither policy appears to have had any association with long-term emission reductions. The lack of practical effectiveness is contrary to conventional expectations, but is not necessarily unexpected.

It is also important to reconcile apparent discrepancies with earlier quantitative analyses that had prompted the move to carbon taxes at the national level, based on British Columbia's experience [53–55]. Earlier works employed valid methodologies, focusing

on econometric analysis of time-series fuel data. Several important points help to explain differences. The earlier works involving British Columbia: (i) focused on gasoline consumption; (ii) considered shorter periods of time after implementation; and (iii) compared results for British Columbia against aggregate averages for the "rest of Canada". While this latter approach is valid, it involves an entity that actually does not exist. On the other hand, as part of the current work: (i) emissions from all relevant sources are considered, as obtained directly from National Inventory Reports; (ii) longer time periods are considered; and, likely most important, (iii) individual provinces are treated as the discrete entities they represent.

A number of future research directions follow from this paper. The analysis here considered long-term emissions trends, which will continue to change over time. Analysis needs to continue into the future as situations and needs change. Grid decarbonization showed an association with long-term emission reductions, and warrants further investigation, particularly understanding the extent of practical limits. Two policy categories, efficiency and vehicle electrification, could not be included in the analysis, and warrant further investigation—the former in terms of developing suitable metrics to relate energy efficiency efforts to emission reductions, and the latter in terms of tracking progress as electric vehicle numbers grow in order to confirm effectiveness for emission reduction. Three policies showed no association with long-term emission reduction in practice: elevated renewable fuel content and, notably, carbon taxation and cap-and-trade. All three have a valid theoretical basis, and further investigation is warranted to better understand why positive trends were not seen, and how to potentially improve the environmental effectiveness in each case. A specific area of concern for carbon taxation appears to be over-reliance on models with embedded assumptions to predict reductions, as opposed to comparison to actual results. One example of research could involve examining the price elasticity of demand assumptions and better relating this to variability and changes in actual markets. Lastly, regarding carbon taxation, the analysis considered the form of the tax as employed within Canada; however, a variety of changes in form could be considered to improve the environmental effectiveness. One simple example relates to how tax revenues are employed, examining revenue recycling, as used within Canada, compared to using collected revenues to mitigate damages or fund emission reduction activities.

## 6. Conclusions

The analysis in this paper was based on a balanced presentation approach to address a controversial topic. Using balanced presentation led to evaluating emission reductions achieved by individual Canadian provinces over time. This shifted the focus away from the point of controversy—in this case, whether or not carbon taxation is useful—to a broader quantitative question. Simple quantitative analysis was employed to review emission trends, with quantitative metrics to identify and confirm the application of relevant emission reduction policy categories by individual jurisdictions.

The results show trends in GHG emissions over the period from 2005 through 2019 for individual Canadian provinces. Three provinces show consistent downward trends, these being New Brunswick, Nova Scotia, and Ontario, while two provinces show consistent upward trends, these being Saskatchewan and Alberta. The other five provinces show no definite trends and exhibit varying degrees of flatness in terms of emissions profiles, with Quebec, British Columbia, and Manitoba the flattest, in that order.

In considering seven diverse emission reduction policy categories, the results also clarify where successes appear to have been occurring, and also where deficiencies or ambiguities exist. Of direct relevance to the initial controversy that prompted the analysis, neither carbon taxation nor cap-and-trade, both economic instruments, appeared to be associated with any long-term reductions in overall emissions at a sub-national level. Two provinces, British Columbia and Quebec, were confirmed to have consistently employed the policies, respectively. They show the flattest emission profiles of all provinces, with no overall reductions. A lack of reductions in both cases is contrary to conventional expectations.

The use of a balanced presentation approach provides broader insights. A third policy category, namely elevated renewable content in fuels, showed, similar to carbon taxation, no association with long-term reductions in overall emissions, despite being intuitively positive. Employed consistently in Manitoba and Saskatchewan, the two provinces, respectively, show relatively flat and increasing emission trends, with no obvious declines. Two policy categories, energy efficiency and vehicle electrification, involve either inadequate data to allow historical comparison, or implementation that is not yet sufficient in any individual jurisdiction to be able to confirm the potential for emission reductions. Both warrant further consideration.

A policy of encouraging fossil fuel production was not associated with emission reductions. This was most evident in the case of Alberta, with increased production associated with crude oil. This province showed increasing emissions, despite also engaging in grid decarbonization. Fossil fuel production was also relevant, albeit to a lesser extent, in British Columbia, primarily associated with natural gas production. The results suggest that emphasizing fossil fuel production is inconsistent with and counterproductive to reducing emissions.

Of the policies considered, one stood out as being associated with long-term reductions in overall emissions—specifically, grid decarbonization. The policy was quantitatively confirmed as undertaken in all three of the provinces showing reduction trends.

Canada's federal government has continued to emphasize carbon taxation, or cap-and-trade, as primary policy tools to reduce emissions. A great deal is riding on the success of the carbon tax, given expectations for decisively large and rapid emission reductions to be achieved. Not only were no long-term emission reductions observed in the provinces employing these policies at a sub-national level, but when the carbon tax system was finally implemented on a consistent national basis in 2019, overall emissions for the country went up, rather than down. Canada could be left in a vulnerable position, facing an awkward emission reduction deficit. If Canada is to achieve its Paris Agreement target, the country may need to reconsider its policy priorities.

Balanced presentation is certainly not new, but the approach offers promise in a modern context. In dealing with contentious environmental issues, more than anything else, an emphasis on pragmatism is required. Balanced presentation represents one option, helping to reconcile policy quandaries and move forward.

**Funding:** This research received no external funding. Support to assist with the APC was provided by the Freedom with Focus Foundation (Public Policy and Good Governance) and gratefully acknowledged.

**Institutional Review Board Statement:** Not applicable.

**Informed Consent Statement:** Not applicable.

**Data Availability Statement:** All of the data in this research come from public sources that are all openly accessible. Specific links to sources, as employed, are provided in References [25,26,78,80–84].

**Conflicts of Interest:** The author declares no conflict of interest.

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
