# Peer review of "Canada as a Case Study for Balanced Presentation to Address Controversy on Emission Reduction Policies"

_sustainability, doi:10.3390/su13147909_

Round 1
Reviewer 1 Report
- In the introduction part, it would be better to point out the significance/impact of this research and the research gap.
- The results and discussion part is clearly presented.
- I think the author should come up with more detailed implications based on the conclusion.
Author Response
Based primarily on suggestions from other reviewers, the paper was revised to include a much more extensive literature review, with this new section termed “Background and Literature.” A discrete section on literature was not originally included because of the way the template was set out, attempting to match the apparent style. This has been corrected. The added section addresses suggested improvements in: (i) providing better context regarding existing theoretical background and empirical research (first check mark point); and (ii) improving references (fifth check mark point). Regarding the first written comment, changes made to the beginning of the paper also address the comment about the introduction. Regarding the third written comment, changes are also made to better explain implications. Lastly, the paper was separately reviewed to confirm grammar and spelling. This included removing unnecessary “superlative” descriptions, as suggested by other reviewers.
Reviewer 2 Report
The authors should add a literature review section
the conclusions should include the limits of the research and future research directions, theoretical and practical implications with better highlighting the originality of the paper
Lines 496-500 are too coloquial, terms like "quite clear, certainly not new, data employed here are available to all, can be replicated by anyone" should be rephrased with more objective ones and not so vague.
The title also should be rephrased to sound more academic, objective, not in a subjective and coloquial manner.
Author Response
The most important revision has been to include a much more extensive literature review, with this new section termed “Background and Literature.” A discrete section on literature was not included because of the way the template was set out, attempting to match the apparent style.
This has been corrected, addressing the first written comment, and also providing: (i) better context regarding existing theoretical background and empirical research (addressing first check mark point); and (ii) improving references (addressing fifth check mark point), which was indicated as required.
Changes have been incorporated to more fully explain why the approach in the paper was undertaken and how this led to the specific method as employed (addressing second check mark point).
Regarding the second written comment, limitations of the work are included in the Methods section, and future research and implications are presented in the Discussion section.
Regarding the third written comment, the paper was separately reviewed to confirm grammar and spelling. This included removing unnecessary “superlative” descriptions and colloquial phrases, as suggested.
Lastly, regarding the final written comment, the title of the paper has been changed, as suggested, to become more straightforward. An observation by another reviewer was that the paper actually represents a case study, and this is incorporated too, noting that many case study oriented papers have been published in the journal Sustainability.
I note that the title has been altered as requested, but the same time, it is useful to note that there has been a trend, at least within North America, to use titles for academic papers that are more like newspaper “headlines,” in order to attract attention and to appear less “stuffy” and “academic.”
Reviewer 3 Report
General comment:
1.The paper has no grammatical problems, English is quite acceptable. But it needs several improvements so that it can be published as a scientific article. In other words, it looks like a case study to me.
Specific comments:
- In the introduction, the author must present the motivation for this article. What goals to achieve. Also, in the introduction, the structure of the article should be mentioned.
- In my opinion, the literature review should be introduced; as a rule, this section should have two to three pages that encourage the analysis of results.
- In the methodology and methods, the author must present all the sources and descriptions of the methods to be applied.
- Findings are very weak and need further improvement and provide clues for future work and demonstrate policy implications.
Author Response
Regarding the first written comment, the observation that the paper represents a case study is important, and actually is received positively by the author. The title has been altered, as suggested by another reviewer, and explicitly describes the paper as a “case study.” Importantly, for the journal Sustainability it is noted that many case study oriented papers have been published, so such an approach is legitimate.
Regarding the second written comment, the motivation for the article is noted in the Introduction, and described more extensively in the Background and Literature section. The structure of the article is also described more fully in the Introduction.
Regarding the third written comment, the absence of a literature review was noted by other reviewers. The revision includes a much more extensive literature review, with the new section termed “Background and Literature.” A discrete section on literature was not included because of the way the template was set out, attempting to match the apparent style.. The paper now provides more thorough explanations for the approach as employed.
This also addresses: (i) better context regarding existing theoretical background and empirical research (addressing first check mark point), which was indicated as required; and (ii) improving references (addressing fifth check mark point), which was indicated as required. This does make the paper longer, now moving to a total of 22 pages, including 4 pages of cited references, the latter up from 2 pages in the original version. The “Background and Literature” section is approximately 6 pages.
Regarding the fourth written comment, the Method section describes the method as employed more fully, and it does list all the sources of data, which are all publicly available. The data sources are also summarized at the end of the paper in the “Data Availability Statement.”
Regarding the fifth written comment, future research directions and implications are more fully described in the Discussion section.
Regarding the second check mark point (research design, questions, hypotheses and methods), these are more fully explained in the Background and Literature section and the Methods section, noting this was indicated as required.
Regarding the third check mark point (arguments and discussion of findings coherent, balanced and compelling), the implications are outlined more fully in the Discussion section, noting this was indicated as required.
Regarding the fourth check mark point (empirical research, are the results clearly presented), the results are more clearly presented in the Results section, noting this was indicated as required.
Lastly, the paper was separately reviewed to confirm grammar and spelling. This included removing unnecessary “superlative” descriptions, as suggested by other reviewers.
Round 2
Reviewer 3 Report
This second version is better. However, the paper presents some problems of legibility. Therefore, please format the paper according to the rules of this journal.